# Factors Associated with Over-the-Counter Analgesic Overuse among Individuals Experiencing Headache

Maram Alshareef 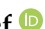

Department of Community Medicine, Pilgrims Health Faculty of Medicine, Umm Al-Qura University, Makkah 21955, Saudi Arabia; mhashareef@uqu.edu.sa

**Abstract:** The prevalence of chronic daily headache (CDH) worldwide is 4–5%. Treatment for CDH with prophylaxis and abortive medications is often delayed, increasing disease severity. Consequently, increased usage of over-the-counter (OTC) analgesics can lead to medication overuse headache (MOH). This study aimed to assess factors affecting OTC analgesic overuse causing headaches. Methodology: A cross-sectional structured survey was conducted using voluntary response sampling. Among 1177 respondents, 459 individuals with headache were enrolled in the study. Results: Most participants were female (73.5%), almost half were aged 20–39 years (48.1%), and over half used OTC analgesics (56%). A significant association was found between OTC analgesic overuse and factors, such as chronic disease ($p = 0.007$), working status ($p = 0.015$), smoking ($p = 0.02$), headache frequency >15 days per month ($p = 0.000$), migraine-type headache ($p = 0.01$), preventive medicine use ($p = 0.018$), and emergency department visit ($p = 0.018$). Conclusion: OTC analgesic overuse among individuals with headache is affected by several factors, including the presence of chronic diseases, working status, smoking, and migraine-type headaches. However, preventive medicine and emergency department visits were considered to have protective effects. Health care providers should screen patients for MOH, increase their awareness, and follow MOH guidelines to treat chronic headache.

**Keywords:** medication overuse; over-the-counter analgesic; chronic daily headache

## 1. Introduction

Chronic daily headache (CDH) is one of the most common conditions observed in a community setting, with a prevalence of 4–5% [1]. CDH is defined as headaches occurring for at least 15 days a month for a period of >3 months. The condition has different treatment approaches depending on the diagnosis of the primary headache disorder. Primary headache disorders causing CDH include chronic migraine, tension type, cluster, or other disorders. CDH needs prophylaxis medicines to be taken daily, in addition to abortive medications that are taken on an as-needed basis for the management of CDH [2].

However, patients may suffer for years without a proper diagnosis [3]. Effective treatment of CDH is often delayed or restricted due to limitations in finding a relevant specialist, reaching the diagnosis, or establishing proper treatment [4]. A major reason for delays in treatment is the long waiting time that can be attributed to the complexity of health care systems and the shortage of pain specialists, further delaying the patient's access to proper chronic pain management [5]. Misdiagnosis of the primary condition by a health care provider is a significant reason, although the probability of receiving an accurate diagnosis by a neurologist or chronic pain specialist is higher than that by other specialists [3,6]. Moreover, the patient's compliance to pain management might diminish over time if they misunderstand the pharmacotherapy or have higher expectations related to chronic pain management [7,8].

Consequently, patients often tend to use over-the-counter (OTC) analgesics, like paracetamol, nonsteroidal anti-inflammatory drugs (NSAIDs), triptan, and opioids, that

are available in different forms OTC without any prescription or can be prescribed by the primary care provider without any restrictions [9,10]. In Saudi Arabia (SA), the prevalence of MOH is 2%, which is comparable with the worldwide prevalence [11]. Access to chronic pain specialists who can provide comprehensive pain management is limited in SA, which can increase a patient's pain despite routine follow-up with neurologists or general practitioners [12,13]. Previous studies conducted in SA have shown that the prevalence of OTC drugs used without prescription was 52.9–81.4% [14,15]. Analgesics were one of the most frequently overused drugs [16]. Not surprisingly, 64.8% of the Saudi population reportedly use OTC analgesics for headaches over a 3-month period [14]. OTC analgesics overuse is defined as the use of analgesics more than two times per week for more than 3 months which unfortunately, can be a cause of MOH, further complicating, overlapping, and masking the original diagnosis of CDH [4,17,18]. Despite the non-safety profile of OTC medication, including liver or kidney toxicity if used more than the maximum daily doses, health care providers and patients continue to misuse them, which can be dangerous [19,20]. Different factors can increase the risk of OTC analgesic overuse, such as unemployment and smoking, which were also found to increase the relapse of MOH [21]. Other associated factors were female sex, low education, lower levels of physical activity, chronic pain, chronic diseases, and frequency of headache attacks [16,22,23].

This study aimed to investigate the factors associated with OTC analgesic use among a population experiencing frequent episodes of headaches as they are more susceptible to have side effects due to overuse, especially refractory headache or MOH.

## 2. Materials and Methods

### 2.1. Study Setting and Sampling

In this study, an online cross-sectional survey was conducted between October 2021 to November 2021 specifically focusing on the Western region of SA. The sampling method was acquired through voluntary responses.

The minimum sample size required for this study was calculated using OpenEpi version 3.0 with the following considerations: The population size is approximately 8,550,000 inhabitants and confidence interval (CI) is set at 95%. The required sample size was calculated to be 385 participants. The inclusion criteria were adults aged >18 years who experienced headaches and used analgesics over the past 3 months. The exclusion criteria were children aged <18 years or those who experienced headaches due to secondary causes such as sinusitis, tumor, or endocrine-related events. This was excluded as a step of data cleaning. If adults were found to have any secondary causes, they were excluded; however, no such adults were identified. Data collection depended on the voluntary response technique.

### 2.2. Study Tool

A structured online MOH screening questionnaire was adopted and modified [24]. The questionnaire was distributed electronically through social media platforms. The questionnaire included details on demographics, chronic diseases, work-related lifestyle factors (if working or not, number of jobs, type of job, and working hours), smoking, sleeping hours, and caffeine intake. In addition, the survey included questions to infer the participant's OTC use and knowledge about the medications used for headaches. Headache Questionnaire is attached in the Supplementary Material S1.

### 2.3. Data Analysis

The data were consolidated in Microsoft Excel spreadsheets and then transferred into the SPSS software (version 22) for statistical analyses. Numerical data is presented as mean ± standard deviation (SD) or as median and range based on the type of distribution for each variable. For categorical variables, data are presented as percentages. The chi-square test was used for categorical values.

For data analysis, SPSS software (version 25) was used. The demographic and clinical data were analyzed using descriptive statistics (frequency, mean, and SD), and the chi-

squared test was used to determine the statistical significance of the association between categorical and nominal variables when *p* value < 0.05.

## 3. Results

Among the 1177 respondents, a total of 459 (age range: 18–65 years) participants experienced headaches and were enrolled in the study. Different factors were assessed to determine the association between OTC analgesic overuse and demographic data. The social factors assessed were age, sex, nationality, marital status, educational level, income, and chronic disease diagnosis. Participants with chronic diseases had a significantly higher number of days of OTC analgesic use during the month than participants without chronic diseases ($p = 0.007$).

No significant difference was observed between the duration of analgesic medication and sex, being in a relationship, education level, and income ($p = 0.116$, 0.219, 0.44, and 0.3, respectively). A stronger association was noted with analgesic use for the following: males, 68 (54.4%); divorced/widow, 31 (58.5%); below secondary educational level, 8 (57.1%); and sufficient income, 131 (50.4%). Table 1 presents the association between sociodemographic characteristics and OTC analgesic overuse.

**Table 1.** The association between sociodemographic characteristics and Over-the-Counter analgesic overuse.

| | | Duration of Using OTC Analgesics | | *p* Value |
| --- | --- | --- | --- | --- |
| | | <3 Months | >3 Months | |
| Age (years) | <39 | 156 (55.3) | 126 (44.7) | 0.204 |
| | 40–59 | 74 (47.4) | 82 (52.6) | |
| | >60 | 8 (38.1) | 13 (61.9) | |
| Sex | Male | 57 (45.6) | 68 (54.4) | 0.116 |
| | Female | 181 (54.2) | 153 (45.8) | |
| Nationality | Saudi | 211 (53.1) | 186 (46.9) | 0.173 |
| | Non-Saudi | 27 (43.5) | 35 (56.5) | |
| Marital status | Single | 96 (55.2) | 78 (44.8) | 0.219 |
| | Married | 120 (51.7) | 112 (48.3) | |
| | Divorced/widow | 22 (41.5) | 31 (58.5) | |
| Educational level | Below secondary | 6 (42.9) | 8 (57.1) | 0.44 |
| | Secondary | 58 (47.9) | 63 (52.1) | |
| | University/above | 174 (52.1) | 150 (46.3) | |
| Income | Sufficient | 129 (49.6) | 131 (50.4) | 0.3 |
| | Insufficient | 109 (54.8) | 90 (45.2) | |
| Have chronic diseases? | Yes | 74 (43.5) | 96 (56.5) | 0.007 * |
| | No | 164 (56.7) | 125 (43.3) | |

\* Chi-square test, $p < 0.05$.

Regarding lifestyle factors associated with OTC analgesic overuse, participants who worked ($n = 133$, 53.4%) used OTC drugs for >3 months compared with participants who did not work ($n = 88$, 41.9%) ($p = 0.015$). Smoking was strongly associated with medication overuse ($n = 55$, 52.4%) ($p = 0.027$), while a lower percentage of non-smokers showed analgesic overuse ($n = 136$, 44.3%) ($p = 0.027$). Although caffeine intake was statistically insignificant for analgesic overuse, a greater proportion ($n = 182$, 48.7%) of individuals consuming caffeinated drinks had headaches than the others ($n = 39$, 45.9%) ($p = 0.065$). Table 2 presents the association between lifestyle factors and OTC analgesic use.

**Table 2.** The association between lifestyle factors and OTC analgesic overuse.

| | | Duration of OTC Analgesic Use | | *p* Value |
| --- | --- | --- | --- | --- |
| | | <3 Months | >3 Months | |
| Do you work? | Yes | 116 (46.6) | 133 (53.4) | 0.015 * |
| | No | 122 (58.1) | 88 (41.9) | |
| Lost your work recently? | Yes | 9 (37.5) | 15 (62.5) | 0.395 |
| | No | 107 (47.6) | 118 (52.4) | |
| Job title | Student | 61 (58.7) | 43 (41.3) | 0.104 |
| | Non-health care worker | 136 (51.7) | 127 (48.3) | |
| | Health care worker | 17 (39.5) | 26 (60.5) | |
| Work more than one job? | Yes | 23 (41.8) | 32 (58.2) | 0.447 |
| | No | 93 (47.9) | 101 (52.1) | |
| Work/study hours daily | <8 h | 105 (50.5) | 103 (49.5) | 0.194 |
| | 8–12 h | 111 (55.8) | 88 (44.2) | |
| | >12 h | 22 (42.3) | 30 (57.7) | |
| Smoking | Smoker | 50 (47.6) | 55 (52.4) | 0.027 * |
| | Non-smoker | 171 (55.7) | 136 (44.3) | |
| | Ex-smoker | 17 (36.2) | 30 (63.8) | |
| Sleeping hours | <7 h | 167 (50.8) | 162 (49.2) | 0.47 |
| | >8 h | 71 (54.6) | 59 (45.4) | |
| Use caffeine containing drinks? | Yes | 192 (51.3) | 182 (48.7) | 0.719 |
| | No | 46 (54.1) | 39 (45.9) | |
| How many cups per day? | 1–2 cups | 125 (53.4) | 109 (46.6) | 0.065 |
| | 3–4 cups | 48 (55.2) | 39 (44.8) | |
| | >4 cups | 19 (36.5) | 33 (63.5) | |

* Chi-square test, $p < 0.05$.

Results showed statistically significant values for OTC analgesic overuse for >3 months for participants who complained of headaches at a frequency of 15–30 days/month ($n = 86$, 66.2%) ($p = 0.000$), migraines ($n = 84$, 58.3%), and cluster headaches ($n = 16$, 66.7%) ($p = 0.001$). Participants who sought different specialists for headache treatment had a significant association with medication overuse ($p = 0.018$), particularly for neurology ($n = 60$, 63.8%) and medicine ($n = 43$, 53.8%) specialists. Table 3 presents the association between headache and OTC analgesic overuse.

**Table 3.** The association between headache and OTC analgesic overuse.

| | | Duration of OTC Analgesic Use | | *p* Value |
| --- | --- | --- | --- | --- |
| | | <3 Months | >3 Months | |
| Duration of headache (years) | <20 | 146 (52.1) | 134 (47.9) | 0.367 |
| | 20–35 | 72 (54.5) | 60 (45.5) | |
| | 35–55 | 19 (45.2) | 23 (54.8) | |
| | >55 | 1 (20.0) | 4 (80.0) | |
| Headache days per month (days) | <15 | 194 (59.0) | 135 (41.0) | 0.000 * |
| | 15–30 | 44 (33.8) | 86 (66.2) | |
| Type of headache | Migraine | 60 (41.7) | 84 (58.3) | 0.001 * |
| | Cluster headache | 8 (33.3) | 16 (66.7) | |
| | Stress headache | 64 (52.5) | 58 (47.5) | |
| | Other | 106 (62.7) | 63 (37.3) | |

**Table 3.** *Cont.*

| | | Duration of OTC Analgesic Use | | *p* Value |
|---|---|---|---|---|
| | | **<3 Months** | **>3 Months** | |
| Visited department for medical consultation | Medicine | 37 (46.3) | 43 (53.8) | 0.018 * |
| | Family medicine | 79 (54.1) | 67 (45.9) | |
| | Neurology | 34 (36.2) | 60 (63.8) | |
| | Emergency | 50 (56.8) | 38 (43.2) | |
| Type of analgesics used during headache | Nonsteroidal anti-inflammatory drugs | 1(100) | 0 (0.0) | 0.135 |
| | Paracetamol group | 198 (54.7) | 164 (45.3) | |
| | Muscle relaxant | 7 (38.9) | 11 (61.1) | |
| | Others | 9 (42.9) | 12 (57.1) | |
| | Paracetamol, codeine, and caffeine combination | 23 (40.4) | 34 (59.6) | |
| Source of analgesics | Physician | 18 (43.9) | 23 (56.1) | 0.131 |
| | Pharmacist | 68 (49.3) | 70 (50.7) | |
| | Family/Friends | 45 (60.0) | 30 (40.0) | |
| | Pursuer | 9 (34.6) | 17 (65.4) | |
| | My own self | 98 (54.7) | 81 (45.3) | |

* Chi-square test, *p* < 0.05.

Interestingly, the lower usage of unprescribed analgesics among participants who tried preventive medication than participants who did not was statistically significant (*p* = 0.018). Although statistically insignificant, more than half of the study participants who took pregabalin, 10 (76.9%); topiramate, 21 (72.4%); and amitriptyline, 9 (56.3%) also used pain relieving medication for >3 months (*p* = 0.199). Notably, they also quit using the medication on the advice of their physician (*n* = 8, 72.7%) or family (*n* = 4, 66.7%). Table 4 presents the association between preventive medicine used for headache and unprescribed analgesic overuse.

**Table 4.** The association between preventive medicine for headaches and unprescribed analgesic overuse.

| | | Duration of Using OTC Analgesics | | *p* Value |
|---|---|---|---|---|
| | | **<3 Months** | **>3 Months** | |
| Ever tried a preventive medicine for headache? | Yes | 68 (43.9) | 87 (56.1) | 0.018 * |
| | No | 170 (55.9) | 134 (44.1) | |
| Type of preventive medicine used | Amitriptyline | 7 (43.8) | 9 (56.3) | 0.199 |
| | Gabapentin | 10 (62.5) | 6 (37.5) | |
| | Topiramate | 8 (27.6) | 21 (72.4) | |
| | Verapamil | 8 (53.3) | 7 (46.7) | |
| | Botox | 5 (50.0) | 5 (50.0) | |
| | Pregabalin | 3 (23.1) | 10 (76.9) | |
| | Others | 22 (44.9) | 27 (55.1) | |
| Still using preventive medicine? | Yes | 41 (41.8) | 57 (58.2) | 0.508 |
| | No | 27 (47.4) | 30 (52.6) | |
| If stopped, why? | Side effects | 13 (52.0) | 12 (48.0) | 0.318 |
| | Fear of addiction | 9 (60.0) | 6 (40.0) | |
| | Family/friend advice | 2 (33.3) | 4 (66.7) | |
| | Doctor advice | 3 (27.3) | 8 (72.7) | |

* Chi-square test, *p* < 0.05.

## 4. Discussion

This study explored factors affecting OTC analgesic overuse causing headaches and found that 2% of the Saudi population is diagnosed with CDH due to MOH, which is comparable to the prevalence worldwide [11]. Although the use of OTC analgesics is acceptable occasionally, to the best of our knowledge, the long-term side effects of OTC medication use have not been reported. However, the overuse of analgesics presents several side effects. Additionally, OTC analgesic overuse can be one of the causes for treatment failure in cases of refractory chronic migraine or tension type headache (TTH) [25].

Several factors can increase patients' intake of OTC drugs. Some of the factors explored in this study were age, sex, marital status, and education level, which were all relevant in the population examined but were not statistically significant. However, a previous study conducted in SA reported age and female sex as significant factors associated with MOH [11]. Although age was not statistically significant, CDH had a high prevalence among the older population. Additionally, a recent systemic review found that age was negatively correlated with chronic headaches and MOH [26,27]. However, it is known that chronic pain and the number of multiple body pain sites are increased in the older population [24]. Interestingly, this observation was supported by another statistically significant finding in this study that showed that the population diagnosed with chronic diseases had an increased incidence of headache and, consequently, OTC drug overuse [16]. Obviously, this can be persuasive, as patients with chronic disease have increased stress, sleep disorders, anxiety, and depression, which can aggravate the headache [27,28].

Among lifestyle and behavioral factors, in addition to smoking, which has been observed in prior studies, work significantly increased the intake of OTC drugs. This result is similar to findings of previous national and international studies, which documented the relationship between these factors and headache [29–31].

Other lifestyle factors investigated were recent work loss, working multiple jobs, sleep deprivation, and caffeine intake. Although these factors had an effect on MOH in some studies, none of these factors had a statistically significant impact on the results [16,22,23].

Meanwhile, this study found that headache characteristics play an important role in OTC analgesic overuse. In the current study, the participants used OTC more when acute headache attacks were noted for <15 days per month, but the use decreased when the headache frequency occurred over a longer duration. Another significant result associated with the use of preventive medication was the decreased OTC analgesic use. In contrast, participants who did not use preventive medications had more OTC overuse than participants who received preventive medications who possibly had fewer headaches. This indicates that the use of preventive medications can reduce the risk of OTC overuse among headache populations [32]. A recent study showed that the use of preventive medication can reduce the number of headache attacks, and hence, OTC analgesic use [33]. Furthermore, recent studies have shown that the general population does not possess adequate knowledge about headaches. Additionally, populations with difficulty accessing medical care tend to use OTC drugs more frequently [4,5].

OTC analgesic overuse was significantly higher in participants who received a clinical diagnosis of migraine, followed by those with tension headaches. This was in contrast with the findings of a previous study that concluded TTH as the most common cause of CDH [11]. This can be attributed to methodology differences, wherein the previous study questionnaire investigated participants' headache symptoms and concluded the diagnosis based on patient description. However, in our study, participants were asked about their diagnosis they received from their health care provider, which could indicate frequent misdiagnosis of headache and ineffective education regarding headache causes and management [5,34]. Interestingly, patients who visited an emergency department showed lower usage of OTC drugs than those who visited other specialty departments for their headaches. Expectedly, the limited prescriptions of medication can reduce the patients' usage of OTC analgesics [35]. Overall, this indicates the importance of the comprehensive treatment of chronic headache through arranging easy access to chronic pain management

centers in the community, which can reduce the waiting time, establish a correct diagnosis, and improve management using an effective treatment regimen. The management must involve and consider an active role for the patient, the use of preventive medication as a treatment for chronic headaches, routine follow-ups, and patient negotiations to overcome obscure factors that might decrease their compliance [4,36].

### 5. Limitations

This study has some limitations. First, it was conducted through personal efforts and in only one geographical area; therefore, the results cannot be generalized. Second, data of participants who voluntarily responded to the online survey were included, which might have introduced a selection bias in the study sample. This study can be replicated through direct contact with the population through phone calls or interview, which requires special funds but could reveal more accurate and in-depth results.

### 6. Conclusions

CDH is a common condition in general practice. One of the causes of CDH is MOH, which occurs due to the misdiagnosis of headaches or a lack of education on OTC analgesics among patients. This type of headache can easily be diagnosed and managed. Establishment of chronic pain clinics in a community setting in SA can significantly improve CDH management outcomes and reduce patient suffering, thus decreasing the burden on secondary and tertiary health care facilities. In this study, several factors were explored, and many factors, such as the presence of chronic diseases, working status, smoking, type of migraine headache, and frequent acute attacks, were found to be predictors of OTC analgesic overuse in the population with headache. While other factors, such as emergency department visits and preventive medicine, had protective effects. The findings of this study may aid health care providers in identifying MOH and suggest that they must examine the medical history of patients with headache in detail to rule out this condition, which is often overlooked. Moreover, diagnosis and treatment of MOH can be performed at outpatient clinics, which would eventually reduce patient suffering and improve overall outcomes.

**Supplementary Materials:** The following supporting information can be downloaded at: https://www.mdpi.com/article/10.3390/clinpract12050074/s1, Supplementary Material S1: Headache Questionnaire.

**Funding:** This research received no external funding.

**Institutional Review Board Statement:** The study was conducted in accordance with the Declaration of Helsinki and was approved by the Institutional Review Board of the Umm Al-Qura University (approval No. HAPO-02-K-012-2021-11-819).

**Informed Consent Statement:** Informed consent was obtained from all subjects involved in the study. Patients signed the cover letter of the questionnaire which stated it would be considered as an agreement for enrollment in the study.

**Data Availability Statement:** Not applicable.

**Acknowledgments:** I would like to thank Bayan Alsharif, King Abdullah Medical City, Makkah Saudi Arabia, for her tremendous effort to analyze the results.

**Conflicts of Interest:** The author declares no conflict of interest.

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
