# Peer review of "Factors Associated with Over-the-Counter Analgesic Overuse among Individuals Experiencing Headache"

_clinpract, doi:10.3390/clinpract12050074_

Round 1

Reviewer 1 Report (New Reviewer)

Line 32-33 the sentence seems out of space. What do prophylaxis medicines and abortive medications do?

Line36-44 are replicates of lines 34-36

Line 50-51 why does lack of the specialist increases the pain?

In line 52 more information is needed. 64.8% is for ever use or last month use.

The aim is not clear for choosing those who experience frequent episodes of headaches. More information is needed for describing conditions specifically applying to those who experience frequent episodes of headache (or why they are important compared to the general population).

One key question is that “frequent episodes” was not emphasized in the data collection process.

Any regression method was considered? What did you do in dealing with the 0 count?

Author Response

Dear Reviewer, thank you for your valuable comments that helped to improve my manuscript.

English editing was done twice for the manuscript through official website 

attached is the response to the comments

Reviewer 2 Report (New Reviewer)

Introduction

 The introduction was mostly well written, covered the main background points and led up to the aim of the study.

 Methods

The authors presented the key elements of the study design and adequately described the used methods. However, some things need to be improved:

2.1. Did you have ethical approval for your study? Informed consent statement? Describe in the methods section. Also, which platform did you use to perform the study?

Although, briefly describe a little bit the process of respondent recruiting because it is missing.

Lines 69-70: Rewrite the last sentence to make it sound more “English” or make it more clear.

2.2 Please provide the questionnaire as a supplementary file.

Results and discussion

 There are more issues within the results section.

Lines 103-104 This sentence does not follow the presented results. Where are presented these average numbers of days?

Table 1 Why did you divide the respondents into these age groups? group <20 has a range of 2 years (18-20)? What is the median age of the respondents?

Lines 106 -110 Which test did you used to get this P values? Please check carefully and describe the statistical methods used in this manuscript because it is not clear what did you actually compared and I’m afraid that you didn’t use the right statistical test to perform this comparison/correlation.

Please add as a footnote the name of the statistical test used to obtain the P values (this applies to all tables in the manuscript)

Also, the author has a character ** that is not visible anywhere else in Table 1. except in the footnote.

Table 4 – Please change the places for columns >3 months and <3 months, because this is the only table in which the results for respondents using OTC analgesics for >3 months are previously presented than those who use OTC analgesics for <3 months. I suggest that the presentation in this table should be the same as in the previous ones, so as not to confuse future readers.

Discussion

LInes 172-173 and many more in this section - You make statements where you emphasize the correlation of some variables with the increase in the use of OTC drugs. It is necessary to distinguish in these conclusions, especially where you refer to your results, is there a correlation between that variable with chronic use (>3 months) or acute use (<3 months) of OTC drugs? In this form, as you wrote in your manuscript just “use” without duration, conclusions do not derive from your results.

Author Response

Dear Reviwer, 

thank you for the valuable comments that help us improve the manuscripts and make them well written, 

attached are the comments, 

regarding the original manuscripts, I could find an attachment section for it so i will send it directly to the assisting editor Bridge, 

Regards

Round 2

Reviewer 1 Report (New Reviewer)

authors have addressed my concerns.

Reviewer 2 Report (New Reviewer)

As authors have revised manuscript according to Reviewers suggestions, l suggest to accept manuscript in the current form.

This manuscript is a resubmission of an earlier submission. The following is a list of the peer review reports and author responses from that submission.

Round 1

Reviewer 1 Report

The authors have performed a pharmaco-epidemiologic investigation during 2 months, to assess the factors affecting OTC analgesic overuse for headache in a region of Saudi Arabia. I am not finding novelty in this work.

Several aspects were found in this manuscript:

-          the current abstract does not directly emphasize the current work, because the authors did not mentioned in the conclusions which are the factors associated with OTC analgesics overuse headache. The abstract should be rewritten, since it is not clear to the reader, as well as highlighting the conclusions of this study.

-          the introduction should be more complete, providing supplementary background in the field, using recent references;

-          the authors should detail the inclusion and exclusion criteria of the study, and how they were applied in the conditions in which it was conducted online. What was the bias, how it has been identified, and overcome ?

-          I am worried about how it was assessed, following the online questionnaire, whether it was a migraine or a cluster headache. The respondents provided a medical diagnosis ?

-          there is a lack of greater comparisons of the results obtained in this experiment with other similar studies, and preferably in the last 5 years.

-          many of the abbreviations are not explained (i.e. ER – table 3, line 199, NSAIDs – table 3, 12.00 – what means ? – table 3, TTH – line 199);

-          at line 8, the CDH should be changed with CHD;

-          the abbreviation SA introduced first time at line 45, should be used al lines 48, 61, etc.

-          the improper use of some English terms and the way the sentences are organized, causes a large part of the transmitted information to be diluted.

-          the authors should carefully spell the text (i.e. line 95 – statical or statistical, etc)

-          English including grammar, style and syntax, should be improved, through the professional help from English Editing Company for Scientific Writings.

Author Response

responses are attached in PDF document

also I would like to attach the English proof reading certificate 

regards

Reviewer 2 Report

The manuscript entitled “Factors Associated with OTC Analgesics Overuse Among Headache Population” should be carefully revised and rewritten. 

At the present paper, author very superficially described factors influencing OTC analgesics overuse, however, this investigation is focused on this issue.

The Methods used (Study tool – social media apps WhatsApp and Twitter) are primitive and not sounding scientific.

Also, the author did not disclose the need for such a study and its relevance.

The current study has additional limitations that unable the author to generalize the results.

I recommend author to correct the aforementioned information and resubmit the paper for review.

Author Response

my responses are in the attachment 

regards

Round 2

Reviewer 1 Report

The authors mostly responded to the comments and suggestions and the manuscript was revised accordingly. I consider it could be accepted for publication in this journal.

Reviewer 2 Report

Dear authors,

Thank you so much for the professional scientific discussion. As I recognized, you have replied to the all comments and remarks; English style was also corrected. Bearing the aforementioned in mind, the paper can be accepted for publication at the present form.